# Developing a Complex Understanding of Physical Activity in Cardiometabolic Disease from Low-to-Middle-Income Countries—A Qualitative Systematic Review with Meta-Synthesis

**DOI:** 10.3390/ijerph182211977

**Published:** 2021-11-15

**Authors:** Martin Heine, Marelise Badenhorst, Chanel van Zyl, Gabriela Lima de Melo Ghisi, Abraham Samuel Babu, John Buckley, Pamela Serón, Karam Turk-Adawi, Wayne Derman

**Affiliations:** 1Institute of Sport and Exercise Medicine, Division of Orthopaedics, Department of Surgical Sciences, Faculty of Health and Medicine, Stellenbosch University, Cape Town 8000, South Africa; chanelkat90@gmail.com (C.v.Z.); ewderman@sun.ac.za (W.D.); 2Sports Performance Research Institute New Zealand (SPRINZ), School of Sport and Recreation, Auckland University of Technology, Auckland 1010, New Zealand; marelise.badenhorst@aut.ac.nz; 3Cardiovascular Prevention and Rehabilitation Program, KITE-Toronto Rehabilitation Institute, University Health Network, 347 Rumsey Road, Toronto, ON M4G 2R6, Canada; Gabriela.MeloGhisi@uhn.ca; 4Department of Physiotherapy, Manipal College of Health Professions, Manipal Academy of Higher Education, Manipal 576104, Karnataka, India; abraham.babu@manipal.edu; 5The School of Allied Health Professions, Keele University, Staffordshire ST5 5AZ, UK; j.buckley@chester.ac.uk; 6Department of Internal Medicine and Center of Excellence CIGES, Facultad de Medicina, Universidad de La Frontera, Temuco 4781176, Chile; pamela.seron@ufrontera.cl; 7Department of Public Health, College of Health Sciences, QU-Health, Qatar University, Doha P.O. Box 2713, Qatar; kadawi@brandeis.edu; 8IOC Research Centre, Cape Town 8000, South Africa

**Keywords:** physical activity, diabetes, cardiovascular disease, metabolic syndrome, qualitative review, systems thinking

## Abstract

Physical activity behaviour is complex, particularly in low-resource settings, while existing behavioural models of physical activity behaviour are often linear and deterministic. The objective of this review was to (i) synthesise the wide scope of factors that affect physical activity and thereby (ii) underpin the complexity of physical activity in low-resource settings through a qualitative meta-synthesis of studies conducted among patients with cardiometabolic disease living in low-to-middle income countries (LMIC). A total of 41 studies were included from 1200 unique citations (up to 15 March 2021). Using a hybrid form of content analysis, unique factors (*n* = 208) that inform physical activity were identified, and, through qualitative meta-synthesis, these codes were aggregated into categories (*n* = 61) and synthesised findings (*n* = 26). An additional five findings were added through deliberation within the review team. Collectively, the 31 synthesised findings highlight the complexity of physical activity behaviour, and the connectedness between person, social context, healthcare system, and built and natural environment. Existing behavioural and ecological models are inadequate in fully understanding physical activity participation in patients with cardiometabolic disease living in LMIC. Future research, building on complexity science and systems thinking, is needed to identify key mechanisms of action applicable to the local context.

## 1. Introduction

Cardiometabolic disease begins with insulin resistance and then progresses to the clinically identifiable high-risk states of metabolic syndrome and prediabetes, before it leads to type 2 diabetes (T2DM) and cardiovascular disease (CVD) [1]. In low-to-middle-income countries (LMICs), the burden attributable to non-communicable disease (including CVD and T2DM) increased from 37.8% of total disability-adjusted life years (DALYs) in 1990 to 66.0% in 2019, with a similar pattern in upper-middle-income countries as well [2]. Cardiometabolic disease imposes a large financial burden on patients and households, while increasing vulnerability to poverty [3]. The co-occurrence of CVD and T2DM further expedites the economic burden in terms of direct medical cost [4] and, arguably, the impact on the physical functioning and societal participation of the patient.

Prevention of cardiometabolic diseases, including T2DM and CVD, includes maintaining a healthy weight, eating healthily, avoiding tobacco use, and being physically active [5]. Countries where the burden of disease is shifting rapidly are struggling to deliver primary and secondary preventative interventions [6]. Public health approaches are failing to address the crucial risk factors (such as physical inactivity) globally [6], while interventions focused on individual lifestyle modifications are largely absent due to intricate and complex resource constraints [7,8,9,10,11]. While high-income countries bear a larger proportion of the economic burden (80% of economic cost), LMICs have a larger proportion of the disease burden (75% of DALYs) [12]. To effectively address the burden of physical inactivity in LMICs, in relation to the increasing burden of cardiometabolic disease, it is imperative that we understand the drivers of physical inactivity (along with the other risk factors), from a primary and secondary preventative point of view. The World Health Organisation (WHO) physical activity and sedentary behaviour guidelines development group argues that there is a specific need for more studies in LMICs that aim to identify how various sociodemographic factors (e.g., age, sex, and socioeconomic status) inform physical activity or modify the health effects of physical activity in an attempt to address global health disparities [13].

Depending on the design, studies may be informed by preconceived conceptual frameworks for behaviour change (e.g., Theory of Planned Behaviour). While such conceptual frameworks have helped to clarify (physical activity) behaviour, they have been criticised for their often linear and phased perceptions of behaviour, which are insensitive to environmental influences [14,15]. Emerging health behaviour models using the Socio-Ecological Framework (which includes social factors, policy, and environmental factors) or Complexity Theory may be more conducive to the complex nature of behaviour [15], particularly in resource-constrained settings. Quantitative methods have been used widely to identify determinants of and factors associated with physical activity. Such studies provide clear quantitative evidence for the relationship between physical activity and a select number of potential determinants (e.g., the relationship between physical activity and built environment). Albeit valuable, these studies may be limited in their scope and comprehensiveness when accounting for the complexity of aspects associated with physical activity within a single study design. 

Alternatively, qualitative studies may provide better insight into the real-world challenges and experiences related to physical activity, unrestricted by prior variable selection. Neither existing qualitative nor quantitative research has been able to fully capture the complex system of physical activity behaviour. However, qualitative research may help to develop an understanding of the people, the practices, and the policies behind the mechanisms and interventions [16]. 

We argue that a broad view across qualitative studies may therefore provide both scope and richness on the complex nature of participation in physical activity. The objective of the present review is, therefore, to obtain a comprehensive, systems-based overview of factors that affect physical activity in patients with cardiometabolic disease (including T2DM and CVD) living in resource-constrained settings, through a systematic review of qualitative studies conducted in LMICs.

## 2. Materials and Methods

### 2.1. Study Design

This systematic qualitative review with qualitative meta-synthesis was conducted in accordance with the guidelines provided by the Joanna Briggs Institute [17]. This review was prospectively registered with PROSPERO (CRD42021241483); however, a formal review protocol was not prepared.

### 2.2. Types of Participants

The included studies were those that addressed physical activity in patients of any age, with cardiometabolic disease (stage 2 and 3 of the staging model provided by Guo et al. 2014) [1], operationalised as studies conducted in patients with (codes refer to ICD 11) insulin resistance (5A44), metabolic syndrome (5A44), pre-diabetes (5A40), cardiovascular disease (11), and type 2 diabetes (5A11) [18]. Studies that solely included patients “at risk” for cardiometabolic disease (e.g., obesity) were excluded.

### 2.3. Context

Studies conducted in one of the 133 low-income, lower-middle-income, or upper-middle-income countries, based on a country’s Gross National Income (GNI) classification for 2021 (representing GNI in 2019) [19], were included.

### 2.4. Phenomena of Interest

The studies included in the review focused on “real-life”, physical (in)activity. This comprised leisure physical activity, including structured exercise and sport participation, as well as physical activity related to manual labour (e.g., subsistence farming). Studies that focused on aspects such as self-care, self-management, or disease knowledge were also considered, as these may obtain relevant experiences on the role of physical activity. Studies that focused on the barriers and facilitators to specific physical activity or exercise programs provided through research activities (e.g., randomised controlled trials, feasibility studies) were excluded, unless access to these respective programs was widely available to the study participants and not dependent on participation in a research project (e.g., a community-wide wellness program).

### 2.5. Types of Studies

Original, peer-reviewed, qualitative studies or mixed-methods studies with a relevant qualitative component, published in English, were included. This included studies with focus groups and interviews, as well as study designs such as phenomenology, ethnography, and community-based participatory research. Studies in which both patients and healthy participants were included, if relevant data could be extracted that pertained specifically to patient experiences. Quantitative studies, document or policy analyses, abstracts, conference presentations, systematic reviews, literature reviews, and commentaries were excluded.

### 2.6. Search Strategy and Data Sources

The full search strategy is available as Appendix A. In short, each search comprised four key blocks: qualitative research AND low-to-middle-income countries AND cardiometabolic disease AND physical activity. We searched the PubMed, Scopus, Web of Science, Cumulative Index for Nursing and Allied Health Literature (CINAHL), SPORTDiscus, and AfricaWide databases (up to 15 March 2021). All avenues were explored to obtain full-text articles, including reaching out to corresponding authors when applicable.

### 2.7. Study Selection

Titles, abstracts (phase 1), and full-text articles (phase 2) were screened and reviewed in CADIMA by two independent reviewers (MH, and research assistant) using predetermined inclusion and exclusion criteria [20]. Any disagreements during the selection process were resolved through consensus, or by consulting a third reviewer where necessary. 

### 2.8. Risk of Bias

Risk of bias was assessed using the JBI critical appraisal checklist for qualitative research [17]. This checklist includes ten questions to determine whether there is congruity between the research methodology with the philosophical perspective, research questions, data collection, representation and analysis of data, and interpretation of results. Risk of bias was assessed by a single reviewer (MH) and independently verified by a second reviewer (CvZ). Any disagreements were resolved through discussion or involvement of a third reviewer where required. 

### 2.9. Data Extraction

Data extraction was conducted based the JBI data extraction tool for qualitative research [17], and included specific details about the population (e.g., disease cluster), context, culture, geographical location, study methods, and phenomena of interest relevant to the experiences and perceptions of patients with cardiometabolic disease. Data extraction was performed by a single reviewer (MH) and verified by a second reviewer (CvZ). Any disagreements were resolved through discussion or involvement of a third reviewer where required.

### 2.10. Data Analysis

A thematic content analysis was performed on the results section of included full-text articles using Atlas.ti software (version 9, Berlin, Germany), using a combination of in vivo and descriptive coding [21,22]. Data considered included the thematic analyses, anecdotes, quotes, tables, and workshop notes, amongst others. The combination of both in vivo and descriptive coding of article findings helped to identify the significance of the text as it was presented, but also allowed the opportunity for a degree of interpretation to grasp the underlying meaning of the information presented [21,22]. Each code (or “finding”) was listed in an evolving codebook (see Appendix A for a condensed version), and supported by example quotes (where possible). Through multiple team discussions (MH, CVZ, MB), the identified codes were then grouped into content categories and “synthesised findings”. All findings were considered in the meta-synthesis, independent of the perceived credibility (unsupported, credible, unequivocal) of the finding [17]. An adapted conceptual framework, based on the socio-ecological model, was used to synthesise findings based on their “proximity” to the patient. Once the core team was satisfied that the analysis and systems map were a genuine representation of the data, these findings were circulated to the broader team for review. The broader review team was then asked to (i) engage with the data and the presentation of the findings, including the systems map based on the interpretation of the findings, and (ii) indicate whether there were aspects not currently contained in the map that the authors proposed to include based on their experience in the field and according to their insights within their own context. To this end, the review team was purposefully composed to ensure that experiences, views, and perceptions from different world regions and genders were represented in the interpretation of the study findings, including South America (PS, GG), North America (GG), Asia (AB), Africa (MH, CvZ, MB, WD), the Middle East (KT-A), and Europe (MH, JB).

## 3. Results

As shown in Figure 1, 1200 unique citations were derived from the various data sources, of which 75 full-text articles were screened for inclusion. A total of 42 articles that met all criteria were included, reporting on 41 unique studies [23,24,25,26,27,28,29,30,31,32,33,34,35,36,37,38,39,40,41,42,43,44,45,46,47,48,49,50,51,52,53,54,55,56,57,58,59,60,61,62,63]. Most studies included patients with either diabetes or pre-diabetes (*n* = 30; 73%), followed by patients with hypertension (*n* = 15; 37%). The 41 included studies were conducted across 22 different countries (see Figure 2), of which four were low-income countries (e.g., Malawi; 10%), 19 were lower-middle-income countries (e.g., India; 46%), and 19 were upper-middle-income countries (e.g., Brazil, South Africa; 46%). The average number of participants (range 7 to 215) included in studies was 30 ± 17 patients, joined by 20 ± 18 stakeholders (e.g., family members, nurses). Not all studies were specifically aimed at physical activity, as some studies focused on illness perception or disease self-management as broader concepts. A full description of each study is available in Appendix A.

### 3.1. Risk of Bias

Appendix A provides an overview of the risk of bias assessment for each included study. Across most studies, there was clear congruity between the research question, methods, analyses, and interpretation of findings. In some cases, there was a lack of clarity (partly due to inadequate reporting) in terms of the alignment between question, methods, and interpretation. The position of the researcher, culturally or theoretically, and the potential impact of the researcher on the participant and vice versa was often not described.

### 3.2. Synthesised Findings

From 208 codes, and through an iterative team process, a set of 61 categories was developed, leading to 26 synthesised findings (Table 1; Appendix A). A systems map of the synthesised findings is presented in Figure 3 and Appendix A. Five additional factors were introduced following engagement with the broader review team (Table 1, Italic).

### 3.3. Intrapersonal

Thirteen synthesised findings describe intrapersonal factors that may promote or hamper the physical activity participation of patients with cardiometabolic disease living in LMICs. A variety of aspects were reported that inform the perceived value of physical activity participation, including medical benefits (e.g., disease control) yet also social and socio-economic benefits (e.g., ability for continued manual labour). Tangible positive experiences from and affinity with physical activity may encourage continued physical activity participation and motivation.

“*I feel healthier, and I also feel my sugar is under control when I exercise.*”(India) [62]

These experiences, in conjunction with knowledge on how to engage in health-promoting physical activity in a safe and effective way (amongst others), as well as awareness of physical activity modalities, may contribute to self-efficacy. 

“*Participants defined physical activity as ‘not sitting in one place’ but ‘keeping busy’. Across most focus group discussions, physical activity was viewed as ‘informal day-to-day activities’ rather than organized exercises.*”(Uganda) [37]

The ability to engage in physical activity may be hampered (or facilitated) by physical health (e.g., comorbidity) and personality traits (e.g., self-discipline, coping). Furthermore, time constraints from work or social responsibilities were reported as barriers to participating in physical activity, thereby limiting capacity. In some cases, challenging life events further reduced capacity (e.g., death of a loved one). The inability to engage in physical activity had direct consequences on the socio-economic stability of the household through loss of revenue, particularly in settings with a reliance on manual labour or farming (i.e., rural areas).

“*Not being able to walk, work, or participate in activities that demanded physical strength was a source of frustration for different reasons. Having a chronic condition implied not being able to share the workload with other family members. This was particularly hard when they could not participate in agricultural activities that are essential for survival.*”(Mozambique, Nepal, Peru) [47]

Conversely, in some settings, physical activity was deemed implicit in day-to-day life, either through manual labour, mode of transportation, or engagement in physical household chores.

“*I work in the farm for 4 h every day. I have no need to exercise.*”(Thailand) [44]

Finally, women were particularly vulnerable to physical inactivity in relation to prevailing family roles, cultural norms and values, or public safety.

“*My father always had dogmatic beliefs and would say that girls shall not be outside of house much. He was against us walking or even going to the gym. We grew up like this.*”(Iran) [57]

### 3.4. Social Environment

Five synthesised findings describe the social context in which the patient engages with physical activity. Social support, either within the direct environment (e.g., family) of the patient or in the wider community (e.g., neighbour, friend), was deemed an important facilitator of or barrier (i.e., lack of social support) to continued physical activity participation.

“*I had walking program with one of my relative for two months, but her husband didn’t allow her to join to me anymore, and I was not motivated to continue. It will be encouraging, if two persons to be along with each other for walking program.*”(Iran) [50]

Conversely, in some cases, social support could be “over done”, resulting in social pressure or scolding. 

“*Just as some participants appreciate family members’ controlling behaviours, some participants resented the constant supervision and reminders to take care of themselves. These reminders were perceived as scolding or nagging behaviours that did not offer alternatives or solutions to challenges and obstacles participants experience in Diabetes self-management.*”(Mozambique, Nepal, Peru) [47]

In some settings, local norms and values with respect to leisure physical activity participation, often in relation to gender-specific expectations, informed physical activity participation or the acceptability thereof. Collectively, these aspects refer to a community of practice in which physical activity is accepted and supported without feeling stigmatised.

“*When people see me ‘walking to exercise’ they often slight me and make derogatory comments that one is greedy and would rather walk long distances than spend money on transportation.*”(Nigeria) [45]

“*I like participation in family walking tours, and it is appropriate for me as I can be with my family.*”(Iran) [57]

An urban environment was specifically referred to as a hindering factor for physical activity (e.g., due to common, sedentary types of employment, access to motorised transport). Conversely, public safety and violence were of concern and an important barrier to physical activity participation, particularly in more urban settings.

Physical inactivity, particularly in sedentary occupations and in urban environment. “*People just don’t walk now-a-days.*”(India) [28]

“*...unsafe parks and pedestrian walkways especially for women…*”(Iran) [50]

### 3.5. Health System and Service Delivery

Three synthesised findings were derived that applied to the level of service delivery. First, the quality and quantity of communication and interaction between patient and provider was deemed critical. Notably, the relationship (e.g., relatability) between the provider and patient had a proposed impact on the perceived quality, relevance, and urgency of the information on physical activity participation relayed to the patient.

“*When the expert or physician that teaches me is of “our” people, I can trust her more and I am more satisfied.*”(Iran) [57]

It was commonly reported that the most accessible form of physical activity was adapting (e.g., quantity, intensity) day-to-day chores or tasks. Independent of the type of physical activity, it was deemed important that the information or service delivery was person-centred. For instance, patients reported information provided on physical activity being too generic and lacking specificity to specific comorbidities, fitness levels, or circumstances. Incorporating the evaluation of physical activity into routine evaluations could facilitate more tailored and sustainable physical activity participation, while diversity in the types of physical activity available could promote person-centred care, access, and sustainability. Communication around the types of physical activities available to the patients was reported as pivotal.

“*The doctor just said I should exercise but did not explain what kind of exercise I should do considering my arthritis problem.*”(Iran) [57]

Based on discussion within the broader review team, an additional three findings, which did not explicitly transpire from the data, were added to this layer. Firstly, out-of-pocket expenses may limit the variety of physical activity types accessible to the patient (e.g., clinic-based secondary prevention programs, gym membership, affordability of exercise equipment), as well as access to routine medical management, thereby potentially affecting health and safety. Most of the literature has focused on the cost benefits of increasing physical activity in reducing the economic burden of cardiometabolic disease [12].

“*I walk that does not have any cost instead of going to the gym.*”(Iran) [57]

Secondly, scarcity in terms of the availability and competencies of healthcare professionals in low-resource settings was highlighted as a potential factor in the quality and/or quantity of advice regarding physical activity that was provided [10]. Furthermore, the quality of medical management may affect a patient’s intrapersonal factors such that it hampers or promotes physical activity participation. Differences in public versus private sector healthcare services may further compound health disparities that affect physical activity participation. Thirdly, public health campaigns targeted at lifestyle behaviours and healthy living, possibly reflecting local health policies [64], may create awareness and promote health-seeking behaviour. 

### 3.6. Built Environment

The built environment (four synthesised findings) had an important role in accessing physical activity modalities, either with respect to proximity, density, or accessibility. Neighbourhood walkability (e.g., quality of roads, safety, adequate lighting), or the lack thereof, was reported commonly as an important factor in physical activity participation (either leisure, or as part of daily commute). 

“*Well, I would like to walk every day for more than 30 min, but the roads in my area are not suitable for walking, there is no walkway or park nearby and I am ashamed of doing any exercise in my home.*”(Bangladesh) [33]

Environmental pollution (e.g., air pollution, or waste) affected outdoor physical activity participation or a context being conducive to safe physical activity. A lack of dedicated (e.g., gym, swimming pool) or non-dedicated public facilities (e.g., parks) conducive to and safe for physical activity was considered a common barrier to physical activity participation.

“*There were only few parks, or other recreational spots where citizens can walk, jog, or exercise in a safe, healthy, and pollution free environment.*”(India) [62]

An urban setting may be a barrier to physical activity participation, whereas a rural environment may be a facilitator—for instance, due to prevailing types of employment or specific geographical features (e.g., hillside, mountains, fertile land for farming). As these features (urban, rural) presented more between studies than within studies, urban and rural disparities did not reflect in the content analysis per se. Hence, an urban/rural finding was added based on discussions within the review team. Furthermore, the review team noted that, specifically in low-resource settings and very much in line with adequate human resources, adequate community-based primary care facilities are paramount in medical management, risk factor identification, continuity of care, and personalised care, amongst others.

### 3.7. Natural Context

Finally, the natural environment (one synthesised finding) played a role in physical activity participation. Participants reported a variety of weather conditions that hampered physical activity participation (e.g., rain, hot weather, cold weather). Subsequently, weather and context (e.g., urban versus rural) were associated with aspects such as air pollution, but also neighbourhood walkability (e.g., muddy roads, street lighting, or crowded streets). 

“*Cold weather makes me not to do exercise.*”(Iran) [60]

“*Women reported that it was considered inappropriate for them to walk on muddy roads and that they were afraid of slipping.*”(Bangladesh) [41]

## 4. Discussion

This systematic review of qualitative studies illustrates a unique perspective on the complexity of physical activity participation for people with cardiometabolic disease living in LMICs. Informed by the socio-ecological model, and following a rigorous systematic approach, 26 findings were synthesised from 41 qualitative studies, conducted in 22 different countries, which could be stratified into factors related to the individual, social environment, health system and service delivery, built environment, and natural context.

In the literature, behavioural change theories such as the Social Cognitive Theory (SCT), the Theory of Planned Behaviour (TPB), Self-Determination Theory (SDT), and the Transtheoretical Model (TTM) have been dominant approaches in understanding the determinants and correlates of physical activity [14,15]. These theories have generally viewed change as a linear, deterministic process based on the interaction of cognitive factors such as knowledge, intention, attitudes, beliefs, and efficacy and intention [65]. Although the utilisation of these theories has informed our understanding of the psychological factors and mechanisms that influence physical activity behaviour [14], physical inactivity remains one of the most important health problems of our time [66]. It has become clear that behaviour, and behaviour change, are a complex phenomenon, influenced by multiple factors [14]. In this sense, socio-ecological models of health behaviour that focus on individual, social, policy, and environmental-related factors may be particularly useful in aiding our understanding of physical activity. As a complex system, a socio-ecological framework sees behaviour as the result of direct, indirect, and interactive influences from factors of multiple levels of the system [15]. Similarly, the findings of this study point toward the multiple interactions, across multiple levels of the person’s ecological system, contributing to an environment (both internal and external to the individual) that either enables or restricts physical activity participation. In line with a systems thinking approach, physical activity behaviour may be influenced by an almost infinite combination of barriers and facilitators [65]. However, the identification of recurrent patterns may be used to develop targeted interventions.

Throughout this review, there were several factors that, in quantitative research, could be classified as effect modifiers and/or confounders yet which were challenging to account for in this qualitative meta-synthesis. Some transpired more explicitly, such as gender, whereas others were less tangible, such as temporal aspects or “geographical context”. With respect to the temporal nature of physical activity behaviour, people would describe a social and physical upbringing in which physical inactivity was implicit—cumulative exposure to various risk factors in conjunction with a potential epigenetic predisposition [67,68]. Geographically, barriers such as safety/violence, air pollution, neighbourhood walkability, and access to physical activity programs appear more prevalent factors in urban settings [69,70]. Conversely, the role of manual labour and subsistence farming in rural settings may affect the relative (perceived) value of physical activity in risk reduction or secondary prevention. Hence, in particularly in rural areas, the role of physical activity in the primary and secondary prevention of cardiometabolic disease may not be so explicit, and other risk factors may be more prevalent [71,72,73]. The impact of changing context (e.g., urbanisation) on physical activity did not reflect explicitly in the factors identified, despite compelling evidence that, for instance, urbanisation or migration impact physical activity participation [74,75,76]. The impact of time has not been fully captured in any of the prevailing models of behaviour [14]. Finally, women appeared more at risk for physical inactivity (particularly in relation to prevailing family roles impacting employment and power dynamics) and appeared to report more barriers to physical activity in relation to safety, cultural or religious norms, and stigmatisation. In this light, there may be a case for a gender-specific approach in addressing physical activity in contexts where this is applicable [77]. 

This review focused specifically on studies in patients with cardiometabolic disease due to its association with “lifestyle”-related risk factors including physical activity [78]. However, the findings of this review might be extrapolated to physical activity participation in the wider population (e.g., community of practice, access to facilities, cultural and religious norms). Within the socio-ecological model, these “pathology-transcending” factors would predominantly be found beyond the intrapersonal level (e.g., access to facilities, community of practice, built environment). At the intrapersonal level, though, disease-specific consideration may be required, including factors such as knowledge, self-efficacy, and physical well-being (e.g., comorbidity). Collectively, the findings highlight and argue for a holistic, intersectoral, approach while also emphasising the specific individual needs in relation to cardiometabolic disease. As with many “wicked” problems [79], each of the factors identified should be recognised as equally necessary, yet equally inadequate on their own [80]. 

### 4.1. Implications for Future Research

This review highlights that physical (in)activity, as a risk factor in and for cardiometabolic disease in low-resource settings, cannot be solely explained by relying on behavioural models. The factors synthesised in this qualitative review, drawn from real-life experiences across 22 different LMICs, highlight that future research in such settings should approach physical activity from a more holistic perspective, including factors related to the person, social context, health system, and built and natural environment. This argument has implications for both research and clinical practice, including how we assess physical activity as a risk factor, whether in clinical practice (i.e., relevance of social determinants) or in epidemiological studies, as well as understanding the feasibility and cost-effectiveness of various interventions to address cardiometabolic health or physical inactivity. Systems thinking and complexity science provide the methodological tools for the development and co-creation of contextually appropriate and targeted solutions; the presented findings may provide an initial holistic entry point for these participatory research activities [81]. Furthermore, the findings of this review provide a potential framework for the development physical activity measures, based on complexity science and systems thinking, to assess risk factors for physical inactivity in low-resource settings. Access to objective measures for physical activity in low-resource settings remains scarce, particularly in routine clinical care, while there are also continued concerns on the (cultural) validity and theoretical underpinning of existing self-report measures of physical activity [82], which may be compounded in low-resource settings specifically [10].

### 4.2. Limitations

This review has some limitations. First, three layers of interpretation took place to obtain the synthesised findings from the perspective of the participant, subsequently the study author/interviewer, and finally during this review process. Consequently, some of the nuances may have been lost across these levels of interpretation. Nonetheless, the aim of a meta-synthesis is to translate existing qualitative research into findings that move beyond the results of primary studies to reach enhanced understandings about the phenomena under review [83]. As such, we placed importance on a rigorous, iterative analysis process that satisfied the research team’s judgement of findings being transparent, comprehensive, and supported by the data [84]. Second, all authors of this review work in a medical and academic setting, albeit in different world regions. Our collective gaze on physical activity in low-resource settings may have affected our interpretation of the codes and partly informed the categories and synthesised findings. A concerted effort was made to ensure that the review team consisted of professionals with a wide geographical background, particularly from LMICs, to ensure the rigor and trustworthiness of our findings and interpretation beyond a specific world region. Restricting inclusion to the English language, and excluding databases such as LILACS and Scielo, may have affected the identification of articles relevant for specific world regions. However, additional searches in these two databases (PS) are not indicative of a large body of literature that was excluded because of this limitation. Finally, it is difficult to ascertain to what extent the review findings are transferable beyond patients with cardiometabolic disease living in LMICs and this should therefore be considered when interacting with this review from the viewpoint of other medical conditions. Future research should explore the value of this conceptual map in other populations or contexts.

## 5. Conclusions

Existing behavioural and ecological models are inadequate in fully understanding physical activity participation in patients with cardiometabolic disease living in LMICs. Future research, building on complexity science and systems thinking, is needed to identify key mechanisms of action applicable to the local context. This review may provide a platform to further develop systems thinking in this field and assist in the conceptualisation of holistic tools to assess such complexity in resource-limited settings.

## Figures and Tables

**Figure 1 ijerph-18-11977-f001:**
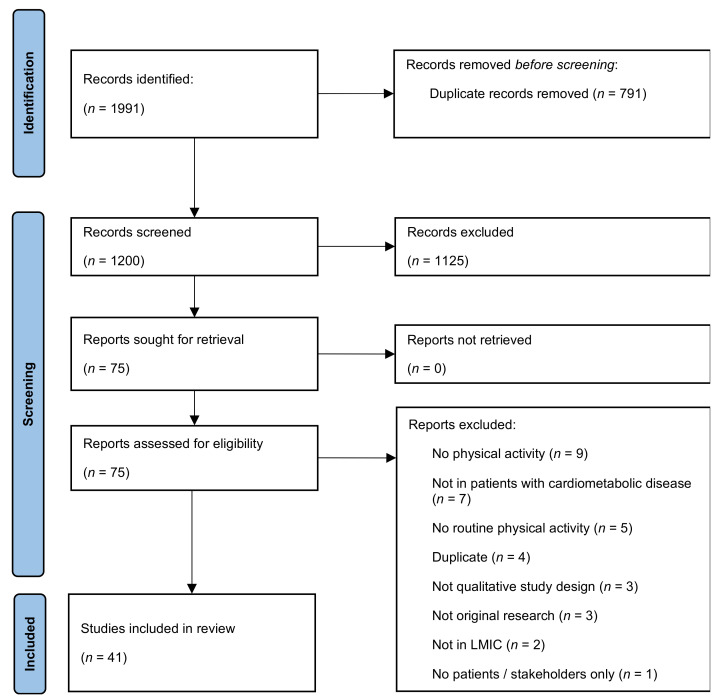
PRISMA flowchart. No studies were identified through other sources (e.g., citation screening). A total of 42 articles met inclusion criteria, reporting on 41 unique studies.

**Figure 2 ijerph-18-11977-f002:**
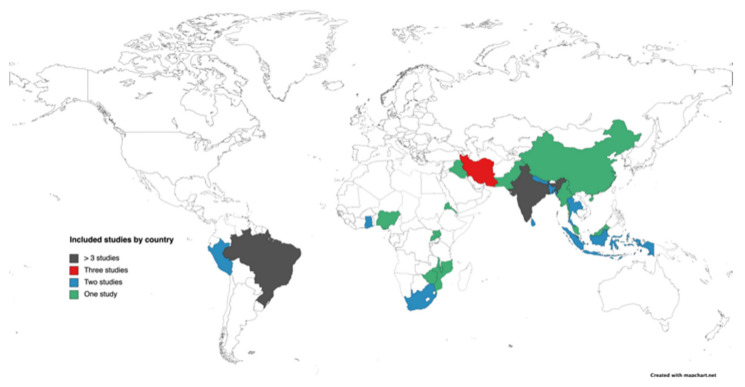
Overview of the included studies by geographical representation.

**Figure 3 ijerph-18-11977-f003:**
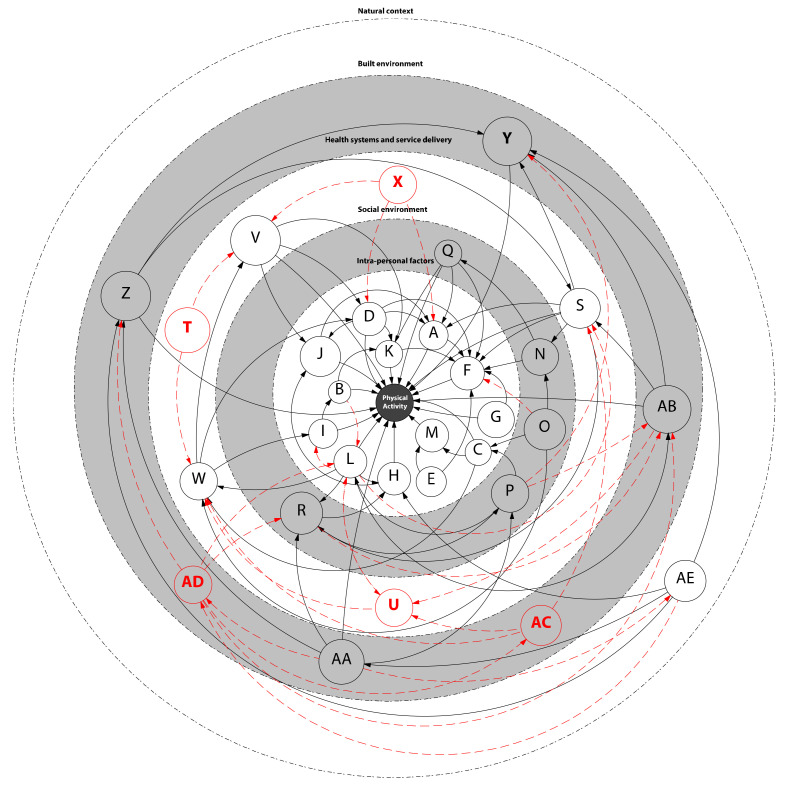
Systems map illustrating the complex nature of physical activity. The arrows indicate mechanisms through which various factors influence physical activity as informed by the underlying qualitative data (grey scale) or as identified within the multinational review team (red). Not all possible relations are shown, most relations will be bi-directional, and all factors are related to each other to some degree. The letters (e.g., A, B, AE, etc.) can be cross-referenced to the synthesised findings presented in Table 1. A high-resolution version of this figure, in which findings are presented in writing rather than coded, can be found in Appendix A.

**Table 1 ijerph-18-11977-t001:** Description of the synthesised findings (*n* = 26 + 5).

Layer	Finding	Description	Figure *
Intrapersonal	Awareness	Awareness of the types of physical activity available, including activity programs on offer or types of physical activity that do not require physical activity facilities (e.g., walking, cycling).	A
	Capacity	Tangible (e.g., equipment) and indirect resources (e.g., conflicting roles and family responsibilities) available to the person to engage in physical activity.	B
	Gender	Characteristics of women, men, girls, and boys that are socially constructed.	C
	Knowledge	Knowledge and understanding of the potential benefits of physical activity in relation to one’s health.	D
	Life events	Isolated experiences that disturb an individual’s usual activities, causing a substantial change or re-adjustment.	E
	Motivation for physical activity	The drive to engage in physical activity; can be informed by a variety of “forces”, either biological, emotional, social, or cognitive.	F
	Personality traits	People’s characteristic patterns of thought, generally stable across time and context. In relation to physical activity, this may include aspects such as acceptance, self-discipline, and coping with life and stress.	G
	Physical activity implicit to day-to-day activities	Physical activity is not an optional behaviour but directly informed by the local context. For example, walking is the only mode of transport available, or physical activity is related to a person’s roles and responsibilities (e.g., household chores, manual labour, farming).	H
	Physical well-being	A person’s physical health and well-being, including exercise capacity or fitness, comorbidity, impairment, or adverse effects in response to being physically active (e.g., fatigue, pain).	I
	Recognition of the value of physical activity	Recognising the potential benefits of being physically active or negative consequences of being inactive in relation to perceptions or experiences.	J
	Self-efficacy	An individual’s belief in his or her capacity or capability to participate in physical activity.	K
	Socio-economic well-being	Having present and future financial security; includes the ability to consistently meet basic needs, make informed economic choices, and maintain financial security over time.	L
	Time	Available time or lack thereof (e.g., time poverty).	M
Social environment	Community of practice	Communities of practice refer to groups of people that share a passion for or affinity with physical activity in general, or certain forms of physical activities. Within these communities, being physically active is accepted and supported.	N
	Cultural and religious norms and values	Cultural or religious values are abstract concepts that certain kinds of behaviours are good, right, ethical, moral, and therefore desirable. Conversely, cultural or religious norms are a standard of behaviour agreed to by respective context. Each set of norms and values may affect physical activity positively or negatively—for instance, due to the acceptance of exercise, family hierarchy, societal roles, and responsibilities.	O
	Public safety or violence	A community in which people can be physically active while safeguarded from crime, disaster, or other potential dangers and threats.	P
	Social support	A support (e.g., friends, family) structure to turn to in times of need or crisis. Social support for physical activity can be of the emotional (e.g., encouragement), instrumental (e.g., equipment), or informational type (e.g., advice).	Q
	Urban environment hinders physical activity	A human settlement with a high population density and infrastructure of built environment limits PA through factors such as prevailing types of employment, access to “inactive” means of transport (e.g., car, taxi, bus). Arguably, an urban environment spans both the social fabric and the built environment, and may partially be informed by aspects related to the natural environment.	R
Health system and service delivery	Diversity in physical activity offering	The scope of formal and informal means of being physically active available to the person, including activity types (e.g., dancing), time and delivery model.	S
	*Availability and competencies of healthcare professionals*	*Availability, competencies (i.e., knowledge), and diversity (e.g., dedicated team for non-pharmacological secondary prevention) in healthcare professionals that are available (including time) to the patient.*	*T*
	*Out of pocket expenses*	*Cost for accessing healthcare services that are not covered by health insurance (when applicable). Such costs may include cost for pharmacological management, access to physical activity programs or facilities, but also cost for travel, absence from paid or unpaid work.*	*U*
	Patient–provider interaction and communication	The quality and quantity of communication between the patient and healthcare system in relation to physical activity and health.	V
	Person-centred care	The care that is provided is tailored to the patient’s context (e.g., disposable income, cultural background) and health status (e.g., comorbidity, health literacy), including recommendations for physical activity and exercise.	W
	*Public campaigns/awareness*	*Community- or population-wide campaigns aimed at improving knowledge, awareness, or behaviour in relation to the value of physical activity and cardiometabolic health.*	*X*
Built environment	Access	The interaction between built environment (transport, proximity) and access to physical activity facilities or modalities.	Y
	Dedicated facilities for physical activity/exercise	Facilities purposed for physical activity or exercise, such as an exercise gym or sports facility.	Z
	Environmental pollution	The introduction of harmful materials into the environment, including air pollution but also pollution due to (plastic) waste or open sewerage.	AA
	Public facilities for physical activity/exercise	Facilities that are available to the wider population without restrictions, such as walkways, green space, and parks. Although dedicated and public facilities are split findings, there could be some overlap (i.e., dedicated facilities for PA available to the public).	AB
	*Primary care facilities/community-based clinics*	*Access to primary/community-based health services is paramount for the medical and risk factor management of cardiometabolic disease, through which disease- or patient-specific physical activity programs may be offered, and to ensure adequate follow-up, amongst others.*	*AC*
	*Urban/rural*	*Some factors were more likely to transpire in an urban area, defined as areas with a high density of human structures (e.g., houses, commercial buildings, health facilities), while other factors transpired more in rural areas (e.g., manual labour).*	*AD*
Natural context	Natural environment	All living and non-living entities occurring naturally. Includes geographical features such as mountains, sea, or desert, as well as aspects related to, for instance, weather or seasons.	AE

A detailed overview of the underlying categories and codes can be found in Appendix A. Factors (*n* = 5) that did not transpire explicitly from the coding yet did during engagement with the findings within the review team are presented in *italic*. * Code (A tot AE) refers to positioning of each finding with the complex systems map/Figure 3.

## Data Availability

All data generated or analysed during this study are included in this published article (and its Appendix A).

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
