# Peer review of "Developing a Complex Understanding of Physical Activity in Cardiometabolic Disease from Low-to-Middle-Income Countries—A Qualitative Systematic Review with Meta-Synthesis"

_ijerph, 2021, doi:10.3390/ijerph182211977_

Round 1

Reviewer 1 Report

An excellent manuscript that was a pleasure to read. This work provides a valuable contribution to the field of physical (in)activity in LMICs. Specifically, it furthers our understanding of the complex interplay between the myriad factors that influence this behaviour, and thus provides a framework or lens for future work. 

Reviewer 2 Report

The Article “Developing a complex understanding of physical activity in cardiometabolic disease from low-to-middle income countries; an “out of the box” qualitative systematic review with meta-synthesis” requires several changes before it will be published. The major strength of the study is this study practical application, and it is relevant for real situation. However, there are some remarks concerning this article:

  1. In my opinion, the title of the article has to be shortened and presented without mistakes (e.g., why used semicolon there is not clear). Additionally, dot is not appropriate at the end of the title. There is no any explanation what is “out of the box” in this case means?
  2. There are to many keywords and some of them are repetitive.
  3. There is not appropriate citation used in whole article (dot’s or comma’s presented before reference) (e.g., 48, 50 lines).
  4. In the Introduction part I would suggest to expand the theoretical background, especially aiming to introduce different theories and prove why it is important to do qualitative systematic review in low-to-middle income countries?
  5. Links to the tables presented below the figures and tables (e.g., line 187 and 191).
  6. In figure 1 presented flowchart of included articles in the research and its description below the figure, but there is no explanation why in the study there taken 42 articles and 41 studies? In figure there is some “snowflakes” (e.g., “*” “**”), but they are not explained neither below the figure in notes nor in the text. Additionally, the number of references presented to these unique studies are different (38 but not 41 or 42).
  7. Figure 3 presented not clear – the text in the figure could not readable at all. I would suggest to use abbreviations and present below the table detailed notes. Moreover, it is not clear why this figure presented at the end of discussion part.
  8. Table 1 is too long. Therefore, I would suggest to divide it into 5 different once according to the layers. Additionally, table title has to be corrected and the text in tables presented in single line format.

Before publication, in my opinion, article must be improved.

Reviewer 3 Report

In the last years, bibliographic reviews are current in science, moreover, they are also necessary for science.

The paper achieves its intended objectives. Through methodology Prisma synthesizes the range of factors that affect physical activity and
therefore, it demonstrates the complexity of physical activity and its specificity in low-resource settings. It focuses on the meta-synthesis analysis of studies conducted in patients with the cardiometabolic disease living in middle-income countries.

The paper presents Prisma methodology. The authors know the technique, its application, and its limitations. Provides important conclusions for future articles on the subject. I consider it to be a good review article.

Round 2

Reviewer 2 Report

The authors did sufficient corrections according to my presented comments and suggestions. I would agree to let this article for publishing, just there is still 2 points to mention for the authors:

  1.    There still some incorrections for the citation (e.g., 146, 154, 465 etc. lines).
  2.    Additionally, the text in the table could be corrected as well (e.g., single space between lines, corrected columns width). 

Author Response

Dear Reviewer,

Thank you for your quick feedback on our revision. We've provided an itemised response below.

1)  There still some incorrections for the citation (e.g., 146, 154, 465 etc. lines).

Response: Apologies for the oversight. We've corrected the three occassions in line 146, 154, 465; as well as a few others we found after carefully scrutinising the manuscript once more.

2) Additionally, the text in the table could be corrected as well (e.g., single space between lines, corrected columns width). 

Response: The table formatting has been done by the editorial office between the original submission and sending it out for peer-review. The document template does not allow me to manipulate the line spacing within the table. I'll ensure that at the proofing stage (once accepted for publication) we'll work with the copy editor to optimise the table formatting to promote optimal use of space while promoting readability.